# Prevalence of Atypical Bacteria in Patients from Different Paediatric Age Groups Diagnosed with a Respiratory Disease

**DOI:** 10.3390/microorganisms12112328

**Published:** 2024-11-15

**Authors:** Cesar Arellano-Arellano, Graciela Villeda-Gabriel, Francisco-Javier Prado-Galbarro, Paola Alejandra Barrientos González, Magali Reyes Apodaca, Uriel Gomez-Ramirez, Dina Villanueva García, Araceli Contreras-Rodríguez, Ma. Guadalupe Aguilera-Arreola, María Isabel Franco Hernández, Israel Parra-Ortega, Alma Lidia Olivares-Cervantes, Norma Velázquez-Guadarrama

**Affiliations:** 1Laboratorio de Investigación en Microbiología y Resistencia Antimicrobiana, Hospital Infantil de México Federico Gómez, Mexico City 06720, Mexico; carellanoaipn@yahoo.com.mx (C.A.-A.); urielgoramirez93@outlook.es (U.G.-R.); lidia5625@yahoo.com.mx (A.L.O.-C.); 2Departamento de Microbiología, Escuela Nacional de Ciencias Biológicas, Instituto Politécnico Nacional, Mexico City 11350, Mexico.; lupita_aguilera@hotmail.com (M.G.A.-A.); 3Departamento de Inmunología e Infectología, Instituto Nacional de Perinatología “Isidro Espinosa de Los Reyes”, Mexico City 11000, Mexico; gracielavilleda@yahoo.com.mx; 4Dirección de Investigación, Hospital Infantil de México Federico Gómez, Mexico City 06720, Mexico; frjavipg@gmail.com; 5Departamento de Neonatología, Hospital Infantil de México Federico Gómez, Mexico City 06720, Mexico; pabg94@gmail.com (P.A.B.G.); dinavgi21@yahoo.com (D.V.G.); 6Unidad de Investigación y Diagnóstico en Nefrología y Metabolismo Mineral Óseo, Hospital Infantil de México Federico Gómez, Mexico City 06720, Mexico; maghimfg@gmail.com; 7Laboratorio Clínico, Hospital Infantil de México Federico Gómez, Mexico City 06720, Mexico; isafrah1967@gmail.com (M.I.F.H.); i_parra29@hotmail.com (I.P.-O.)

**Keywords:** end-point PCR, prevalence, mixed infections, RDS, pneumoniae, atypical bacteria

## Abstract

Atypical bacterial pathogens present the ability to induce pulmonary damage. At present, there are no available phenotypic diagnosis tests that achieve up to 100% reliability. Therefore, clinicians must utilise molecular techniques for the detection and identification of these pathogens. The main objective of this research was to evaluate the prevalence of atypical bacteria in paediatric patients from different age groups. A total of 609 clinical samples were collected from paediatric patients who presented with an adverse respiratory condition during the period from March 2021 to February 2024. DNA was extracted from the samples, and end-point PCR was performed to detect atypical bacteria. Statistical analyses were performed to evaluate the bacterial prevalence and assess clinical data from newborns and mothers that could be related to RDS. A total of 139 patients exhibited at least one atypical organism (22.82%). *Ureaplasma parvum* was more prevalent in neonates, while *M. pneumoniae* and *C. pneumoniae* were more prevalent in older infants. Atypical bacteria can be present in all seasons of the year, but their prevalence increases during hot weather. Mixed infections due to atypical bacteria may occur. The risk factors related to the development of RDS are prematurity, low weight, and orotracheal intubation.

## 1. Introduction

Atypical bacterial pathogens are difficult to culture. These organisms are obligated/facultative intracellular pathogens that present the ability to induce pulmonary damage. The principal organisms belonging to the atypical bacteria group are mycoplasma, chlamydia, and ureaplasma [1]. Although *Legionella* and *Bordetella* are not considered to be atypical bacteria, their low frequency and clinical importance in causing respiratory damage are also relevant research aspects.

Bacteria belonging to the family *Mycoplasmatacea* (*Mycoplasma hominis*; *Ureaplasma urealyticum*; *U. parvum*) usually cause infections in the urogenital tract of both men and women of reproductive ages. These mycoplasma species gain special relevance in the respiratory tract. During pregnancy, the urogenital infections induced by some of these organisms can be vertically transmitted by ascending from the inferior genital tract and reaching the amniotic liquid, where they multiply before reaching the foetus, infecting its respiratory airways. Other transmission mechanisms have been observed during vaginal birth, where the baby is infected through contact with the infected birth canal [2,3].

Atypical pneumonia (AP) is a respiratory disease involving atypical bacteria, including the species *Mycoplasma pneumoniae*, *Chlamydophila pneumoniae,* and *Legionella pneumophila*. The importance of the identification of these organisms lies in the therapeutical approach, which is entirely different from typical pneumonia (TP). Most atypical bacteria lack a cell wall, conferring intrinsic resistance to β-lactams. Therefore, a differential diagnosis has been proposed based on clinical symptomatology, physical signs, and laboratory data, e.g., bacterial identification [4,5].

*M. pneumoniae* and *C. pneumoniae* are widely associated with countries with high rates of pharyngitis, and they have been detected in patients who have previously been diagnosed with otitis media, common cold, rhinosinusitis, and acute respiratory disease and who have undergone a tonsillectomy. Additionally, both have also been associated with respiratory distress syndrome (RDS); however, *M. pneumoniae* could produce the community-acquired respiratory distress syndrome (CARDS) exotoxin, which has been considered to be homologous to the pertussis toxin [6].

*U. urealyticum*, *U. parvum,* and *M. hominis* have been associated with infectious RDS and other acute and chronic respiratory diseases in newborn subjects, causing morbidity and mortality [7].

Both pneumonia and RDS are completely different diagnoses; however, a histologic finding common to both diseases is direct damage to the lung parenchyma. The presence of bacteria in the tissue is thought to be the cause. However, atypical bacteria induce lung epithelial damage, inducing an increase in surfactant proteins. Surfactant proteins are well known as biomarkers for epithelial damage that is highly observed in respiratory diseases [8].

An early microbiological diagnosis to identify AP-associated bacteria is crucial to provide effective treatment. It is important to mention that mycoplasmas lack a cell wall and are intrinsically resistant to all beta-lactams. Consequently, treatment with penicillin derivatives does not eradicate mycoplasma [9].

Macrolides are the treatment of choice for mycoplasma infections in paediatric patients. Regarding the emergence of resistant strains, fluoroquinolones are used as an alternative treatment [5]. Macrolides and quinolones are also considered to be first-line treatments of pneumonia by *L. pneumophila* and *Bordetella* infections [10,11].

Another respiratory disease in which atypical bacteria can be identified is cystic fibrosis (CF). In CF, the respiratory system is progressively affected, triggering obstruction, chronic inflammation, and the establishment of bacterial infections, mainly due to alterations in the microbiota residing in the respiratory tract [9].

Whooping cough has also been associated with atypical bacteria [12]. This respiratory disease has notably diminished due to the development and application of vaccination schedules; nevertheless, for several reasons, cases are still identified, specifically in paediatric patients. *Bordetella pertussis* has been classified as the main strain responsible for the development of the disease [5].

Currently, no phenotypic diagnosis tests are available that achieve up to 100% reliability. Therefore, clinicians must utilise molecular techniques for the detection and identification of atypical bacteria. Polymerase chain reaction (PCR) is reliable due to its high sensitivity and specificity. Although several real-time-PCR kits have been developed for the identification of these bacteria, these methods are still considered to be highly expensive; additionally, the number of transcripts present in a clinical sample cannot always be attributed to the presence of the disease or to the severity of the infection. Therefore, end-point PCR is still considered to be an affordable and ideal diagnostic method [13].

The main objective of this research was to evaluate the prevalence of atypical bacteria in paediatric patients from different age groups.

## 2. Materials and Methods

### 2.1. Collection of Biological Samples

A total of 609 clinical samples, which included bronchial aspirates (n = 410), tracheal aspirates (n = 20), sputum (n = 45), bronchial alveolar lavages (n = 5), and nasopharyngeal swab (n = 129), were collected from paediatric patients who presented with an adverse respiratory condition. The samples were collected during the period from March 2021 to February 2024 from two main paediatric centres: Hospital Infantil de México Federico Gómez and the Instituto Nacional de Perinatología. Each clinical sample was obtained under informed consent, and the study was previously approved by the Ethics Committee with protocol number HIM/2017/110 SSA. 1434. The patients’ clinical data were obtained from their electronic records. Table 1 describes the classification of the collected samples, considering the age group and the diagnosis of the patients, which included respiratory distress syndrome (RDS), pneumonias, and other respiratory diseases, i.e., altered pulmonary transition, transitory tachypnoea, respiratory insufficiency, respiratory alteration, asthma, bronchitis, and whooping cough syndrome. The classification of the age group was performed according to NOM-008-SSA2-1993 [14] and the clinical diagnosis of the patient (Table 1).

### 2.2. Mucolytic Treatment of the Clinical Samples

The mucolytic treatment of the biological samples was performed as follows: a total of 500 µL of clinical sample was aliquoted in 1.5 mL microcentrifuge sterile tubes. An initial volume = 200 µL N-acetyl cysteine 5% was added to each tube. All tubes were incubated at 37 °C for 30 min. Then, 200 µL NaOH 0.4% was added to each sample. Incubation was performed at room temperature for 30 min. After incubation, 200 µL Tris-HCl 1 M pH 7.0 was added to the sample. All tubes were homogenised with vortexing and subsequently centrifuged at 13,000 rpm for 15 min. Finally, all supernatants were decanted. The pellets were stored for the isolation and purification of the total DNA.

### 2.3. Isolation and Purification of Total DNA

The isolation and purification of the total DNA were performed using a Wizard Genomic DNA Purification Kit (Promega, Madison, CA, USA) according to the manufacturer’s instructions, with slight modifications, using glycogen (Thermos Scientific, Waltham, MA, USA) in isopropanol. The DNA was quantified using the Biotek^®^ EPOCH spectrophotometer (Winooski, VT, USA).

### 2.4. Molecular Identification by End-Point PCR

The molecular identification of atypical pathogens from the clinical samples was performed through PCR reactions in a final volume = 20 μL, which included 10 μL Taq Master Cristal (Jena Bioscience^®^, Jena, Germany) and 10 pmol of each primer (Table 2). The primers were designed with the program Primer 3 plus version 0.4.0 using 50 ng/μL total DNA. All reactions were placed in a thermal cycler (Maxygene II, Therm-1001, Union City, CA, USA) under the following conditions: initial denaturation at 95 °C for 10 min, followed by 35 cycles of denaturation at 95 °C for 30 s, annealing at 61.3 °C for 1 min, and extension at 72 °C for 1 min, followed by a final extension at 72 °C for 5 s. The annealing temperature for the identification of *U. parvum* was modified at 58 °C.

Additional PCR reactions were also performed as a quality control step, which included a positive control for each bacterial pathogen. As these bacteria are uncultured, the positive controls for each bacterium were obtained by cloning specific DNA fragments of *L. pneumophila*, *M. pneumoniae*, *M. hominis*, *C. pneumoniae*, *B. pertussis*, *U. urealyticum*, and *U. parvum* into the vector pJET1.2/blunt (ThermoScientific^®^) and transforming them into cells of *Escherichia coli* DH5α. A control consisting of reagents without DNA and another control consisting of the human constitutive gene β-globin (HBB) were also included. HBB is an internal control of the PCR; it ensures that the extraction is carried out correctly and determines the presence of cells infected by any of the obligate or facultative intracellular bacteria.

The PCR products were then identified through electrophoresis in an agarose matrix at 1%, which was previously added with MIDORI green (Nippon genetics^®^, Tokyo, Japan) at 10%. The conditions for the electrophoresis gel are described as follows: 90 V and 300 mA for 45 min. Amplicons were visualised using the iBright CL1000 imaging system (Thermo-Fisher Scientific, Waltham, MA, USA).

### 2.5. Statistical Analyses

A bivariate analysis was performed by applying the chi-squared test for the evaluation of the association between the dependent variable (RDS) and each of the independent variables defined in this study. Those variables that presented a *p*-value < 0.20 were selected for a multivariate Poisson model with robust variance to estimate the reason for the adjusted prevalence between the dependent and independent variables, with their respective confidence intervals at 95%. Then, the prevalence of atypical bacteria was compared for each season of the year, the type of respiratory disease, and the paediatric group by applying the chi-squared test.

The clinical data from each patient were analysed with STATA (StataCorp, College Station, TX, USA) v.17.0 [15]. A *p*-value < 0.05 was established for statistical significance.

## 3. Results

### 3.1. Frequencies

Both the absolute and relative frequencies of the atypical bacteria were determined in this study. A total of one-hundred-forty samples exhibited at least one atypical organism (22.83%), where one-hundred-thirteen presented one bacterium; twenty-six samples presented two bacteria, and there were three clinical cases in which up to three bacteria were identified. As represented in Figure 1 and Figure 2, *U. urealyticum* was observed in up to eighteen patients (2.95%), while *U. parvum* was identified in fifty-seven patients (9.35%), *M. hominis* in twenty-two patients (3.61%), *L. pneumophila* in three patients (0.49%), *M. pneumoniae* in thirty-nine patients (6.40%), *C. pneumoniae* in twenty-three patients (3.77%), and *B. pertussis* in five patients (0.82%). Although the types of samples present some limitations due to their composition, such as sputum containing a high concentration of mucus or nasopharyngeal exudates that exhibit low cellularity, the use of N-acetyl-cysteine and glycogen made it possible to extract DNA and molecularly identify the atypical bacteria (Figure 1).

### 3.2. Mixed Infection

Mixed infections accounted for up to 16.7% of the total number of clinical cases (Figure 3). In our study, we define a mixed infection as an infectious process where at least two or more atypical pathogens are simultaneously involved in the infection site.

### 3.3. Prevalence by Season

The prevalence of atypical bacteria according to environmental factors, i.e., the four seasons of the year, the type of respiratory disease, and the age group, are plotted in Figure 4, Figure 5 and Figure 6. Most of the pathogens were identified throughout the year, with no statistical significance between the seasons. Only the presence of *M. pneumoniae* was determined as being statistically significant during spring.

### 3.4. Prevalence of Respiratory Disease

The second statistical analysis revealed a significant difference in the identification of *U. parvum* and *M. pneumoniae* in the patients with RDS when compared to other respiratory diseases. A statistically significant difference was also observed in the identification of *L. pneumophila* and *B. pertussis* in those patients diagnosed with pneumonia when compared with other diseases (Figure 5).

### 3.5. Prevalence by Age Group

The third statistical analysis showed a statistically significant difference in the detection of each atypical bacteria regarding the different age groups of the study. Although all the atypical bacteria were statistically significant according to the specific age groups, *U. urealyticum*, *U. parvum*, and *M. hominis* were exclusively identified in the paediatric newborn patients and were found to have significantly high frequencies. However, *M. pneumoniae* and *C. pneumoniae* were the most frequently detected pathogens in those paediatric patients aged less than one month old and were found to be statistically significant in the older infants. Finally, *B. pertussis* and *L. pneumophila* were observed in the young infants.

### 3.6. Risk Factors Related to RDS

A total of 272 clinical history datasets were collected from the newborn group diagnosed with RDS. Table 3 describes the clinical variables associated with the syndrome, while Table 4 presents the adjusted prevalence between RDS (gravity is measured by the amount of oxygen in the blood) and the independent variables of the study.

Gestational age was determined using the following values: extremely premature (<27 weeks of gestation), 40.8%; highly premature (28–31 weeks of gestation), 14.6%; moderately premature (32–33 weeks of gestation); late premature (34–36 weeks of gestation); and full term (≥37–41 weeks of gestation). Weight was classified as adequate (2501–3999 g); macrosomic (≥4000 g); low weight (1501–2499 g); very low weight (1001–1500 g); and extremely low weight (≤1000 g). A total of 23.2% (n = 79) of the newborns in this study were identified as having at least one atypical bacterium.

According to the obtained clinical data from both the mothers and paediatric patients, a bivariate statistical analysis was performed to evaluate the significance between those patients diagnosed with RDS and the patients with no syndrome. Table 3 demonstrates statistically significant differences between the following variables: weight, orotracheal intubation, and prematurity.

Table 4 shows that the variables of ventilatory support, intubation, low weight, and premature birth were positively associated with RDS. If the patient requires ventilatory support, the risk of RDS increases 2.4 times; in addition, if the support is invasive, the risk of developing RDS increases 2.7 times more than if it is not invasive. If the patient is underweight, the risk will be 1.6 times higher than if the patient is not underweight, and, if the patient is premature, the risk increases 3.0 times compared to if the patient is not premature.

### 3.7. Relationship Between Atypical Bacterial Infection and RDS

Finally, no statistical significance was observed among the patients with different RDS severity regarding an atypical bacterial infection (Table 5).

## 4. Discussion

Atypical bacteria play an important role in several respiratory diseases. These organisms are commonly associated with AP. However, other diseases, i.e., RDS, cystic fibrosis, and other adverse respiratory conditions, can also be considered as causes or consequences of an infection induced by these bacterial pathogens.

In our study, it was observed that 22.8% of the clinical samples obtained from those patients diagnosed with a respiratory disease exhibited at least one atypical bacterial species. This finding is highly consistent with other reports worldwide (6–40%) [16,17]. In 2022, the National Institute of Statistics and Geography (INEGI) in Mexico reported that pneumonia is the ninth most common cause of death among men and the sixth most common cause of death among women [18]. It is estimated that one-third of pneumonia cases worldwide are induced by atypical pathogens. We suggest that, in cases of pneumonia in which atypical bacteria are not detected, other types of bacteria or viruses must be involved [19]. In addition, there is also consistency in *M. pneumoniae* being one of the most frequent bacteria detected, as has been observed in other studies [20].

Mixed infection cases were observed. Clinical cases were identified where species of the genus *Ureaplasma* were accompanied by *M. hominis*. Our results are highly consistent with reports from other studies, for example, Tadera et al. (2023) [16]. *Ureaplasma* is considered to be a commensal coloniser of the urogenital tract of up to 53% of the female population worldwide, thus being able to colonise the oropharyngeal mucosa of newborns, resulting in the consistent identification of *U. parvum*, *U. urealyticum*, and *M. hominis* [21]. In eight clinical cases, *M. pneumoniae* and *C. pneumoniae* were identified simultaneously. Both organisms are considered to be the most prevalent in almost all paediatric age groups, except in newborns. This association was also observed by Chaundry et al. in 2022 [22]. However, in 2019, Hagel et al. reported the absence of these atypical bacteria in respiratory tract infections in the same age groups [23]. These findings suggest that atypical bacteria circulate heterogeneously in different regions of the world.

The highest prevalence rate of *U. parvum* was observed exclusively in newborn patients, followed by *M. pneumoniae*, which was present in most paediatric age groups, with greater statistical significance within the older infant group. However, Waites et al. (2017) found a higher proportion of *M. pneumoniae* in school-aged and adolescent groups, while Rueda et al. 2019 observed a higher prevalence of *M. pneumoniae* in preschoolers [6,20].

Atypical bacteria are prevalent throughout the year; therefore, it is important to be vigilant against these bacteria at all times. *M. pneumoniae* was significantly more predominant in spring compared to the other seasons. The highest percentage of positive cases overall was also observed in spring. Some research has determined a correlation between the prevalence of *M. pneumoniae* and warm seasons, i.e., summer and autumn [6], and these data may contradict our results; however, a gradual increase in temperature was observed in Mexico City during the spring season period from 2022 to 2023 [24], justifying the prevalence of *M. pneumoniae* in warm climates. During winter, most cases in the newborn group were attributed to *U. parvum*, while the proportion of different atypical bacteria was determined to be homogeneous during autumn and summer.

Some studies have correlated the presence of mycoplasma with RDS. In this investigation, the results strongly suggest a statistically significantly higher probability of identifying *U. parvum* and *M. pneumoniae* in patients diagnosed with this syndrome compared to patients diagnosed with pneumonia, CF, or another respiratory disease. RDS can be considered a pre-stage to an event of pneumonia or another respiratory disease. Therefore, it is important for physicians to recognise these types of pathogens and their presence in clinical cases of RDS in order to provide a specific treatment before the patient’s respiratory problem worsens [25].

In this research, we evaluated the possible relationship between an infectious process induced by atypical bacteria and the presence of RDS; however, this direct association was ruled out because RDS may occur regardless of whether or not there is an atypical bacterial infection—at least by one of the strains studied here—since there are reports that other species of mycoplasma (*M. fermentans*, *M. genitalium*, *M. penetrans*, and *U. psitacci*) are related to the induction of respiratory tract infections. These facts demonstrate that atypical bacteria are not responsible for RDS and only serve as a factor that can trigger it [26,27].

Bacteria that usually affect the respiratory tract of newborns are described as mycoplasma. These organisms colonise the mother’s urogenital tract. During delivery, the newborn is exposed to pathogens established in the birth canal, allowing these bacteria to colonise the baby’s airways, inducing respiratory distress and even pneumonia [19,21]. Our results demonstrated the predominance of infections caused by *U. parvum*, followed by *M. hominis* and a lower prevalence of *U. urealyticum*, in newborns. Contrary to our study, García et al. (2024) reported a higher rate of clinical cases where both *M. hominis* and *U. urealyticum* were identified compared to cases where *U. parvum* was identified; their study was also carried out using molecular biology [2]. The study conducted by Carrera et al. (2017) [28] identified the prevalence of *M. hominis* and *U. urealyticum* at 3% and 22%, respectively. According to our results, *M. hominis* was identified in 3.61% of the study subjects, while *U. urealyticum* was observed in 2.95% of the samples. However, our research was able to identify atypical pathogens at the species level by considering the genes specific to the *U. parvum* species. Conversely, while the results described by Carrera et al. were obtained through culture and/or PCR techniques for the identification of atypical bacteria, the authors did not fully describe the biochemical tests or the primers used in their study to differentiate between *Ureaplasma* species [28].

The risk of infection does not differ according to the type of delivery. Mycoplasmas are small bacteria (<1 µm) with the ability to cross the placenta and establish contact with the foetus even before birth. This fact demonstrates the lack of statistical significance among patients with RDS who had a natural delivery or caesarean section [23]. Recently, it has been shown that not all mothers who have been previously identified as having a *Ureaplasma* spp. infection transmit the organism to the baby [29,30]. Several studies have reported that mycoplasma causes the premature rupture of membranes (PROM). Mélendez et al. (2020) associated mycoplasma with CCV and PROM. Both parameters, CCV and PROM, were statistically analysed to determine their possible relationship with RDS [31]. As a result, no association was determined. We suggest that CCV and subsequent PROM could be triggered by non-mycoplasma bacterial agents, or by different *Mycoplasma* species than those studied in this research, for example, *M. genitalium*, which has been reported as a bacterium that can cause several disorders in women [32]. Peretz A et al. (2020) reported that genital mycoplasma colonisation can induce complications during pregnancy, including regarding premature birth and among low-birth-weight newborns [33].

In the present study, it was observed that weight, prematurity, and respiratory support in newborns were significantly related to RDS. If the newborn has RDS, ventilatory support will be necessary. On the other hand, weight and prematurity are conditions that should be considered from a developmental point of view: if a newborn is premature, they will be born with a low weight and, consequently, the organs will not develop optimally and the immune system will not mature, making them prone to infections. According to our results, there could be a relationship between these conditions and the presence of mycoplasmas since it is documented that these pathogens affect the development of the foetus and can lead to prematurity [34]. However, premature birth can also be attributed to social, cultural, and biological factors that can induce certain conditions during pregnancy, as asserted by Cernadas et al. (2019) [35].

CF patients, given their condition, are highly susceptible to bacterial infections due to their impaired respiratory microenvironment. It is very common to identify atypical bacteria in the early stages of respiratory dysbiosis. In our study, we were able to identify atypical pathogens in 25.5% of these samples. The predominance was led by *M. pneumoniae*, followed by *C. pneumoniae*. However, other bacteria can be identified as the disease progresses, for example, *S. aureus*, *P. aeruginosa*, and yeasts (e.g., *C. albicans*), which cause the displacement of the atypical bacterial infection. The main focus for these patients is to treat the pathology itself, and the mistake of omitting the diagnosis of infections caused by atypical bacteria can occur. It is important to identify these bacteria in order to complete a specific therapy to treat all types of infection [11].

An interesting part of this research was the observation of cases of *B. pertussis*. Although we were unable to obtain the patients’ medical history to verify their vaccination status, the infections suggest that they were not vaccinated. These paediatric patients belonged to the group of younger infants (3 months [n = 1]; 4 months [n = 2]; >9 months [n = 1]). Similar results were reported by El Basha et al. (2019) in Egypt [36]. Although the vaccine should be administered at 2, 4, 6, and 18 months, the patients were > 2 months old, so we posit, first of all, that these infants had not yet received the first dose; however, as mentioned in the study reported by Ma et al. (2021), the Pa vaccine is safer for the patient but has decreased in its efficacy [37]. This information corroborates the fact that vaccines remain essential for the immunological protection of newborns.

In many parts of the world, empirical prescription is still practised, based mainly on both clinical and radiological diagnoses, leaving aside the identification of the possible causal agents of the respiratory disorder [38]. These practices may represent up to 50% of unidentified pneumonias. According to the findings described in our study, at least 22.8% of the clinical cases analysed could involve the presence of atypical bacteria.

## 5. Conclusions

Infections caused by atypical bacteria are not considered to be seasonal. On the contrary, they can present at any time of the year, increasing their prevalence in hot weather.

The presence of these atypical pathogens represents almost a quarter of the studied cases. This prevalence is important and must be taken into account in the detection and treatment of these microbial agents. The infection is not always exclusively induced by one single pathogen, so mixed infections are possible.

All paediatric age groups appear to be affected by several atypical pathogens, and the prevalence will depend on the geographic location. The exclusive prevalence of *M. hominis*, *U. urealyticum,* and *U. parvum* was observed in newborn patients, while *M. pneumoniae* and *C. pneumoniae*, which were identified in the remaining paediatric groups, were found to be absent in newborns but increasing in prevalence in older infants. The atypical pathogens identified as the most prevalent in the present study were *U. parvum* and *M. pneumoniae*.

No significant relationship was found between infections caused by atypical bacteria and a diagnosis of RDS in newborn patients; however, the infection itself is considered to be a risk factor for the development of RDS, which can increase with other significant risk factors (i.e., prematurity and low weight) in newborns.

## Figures and Tables

**Figure 1 microorganisms-12-02328-f001:**
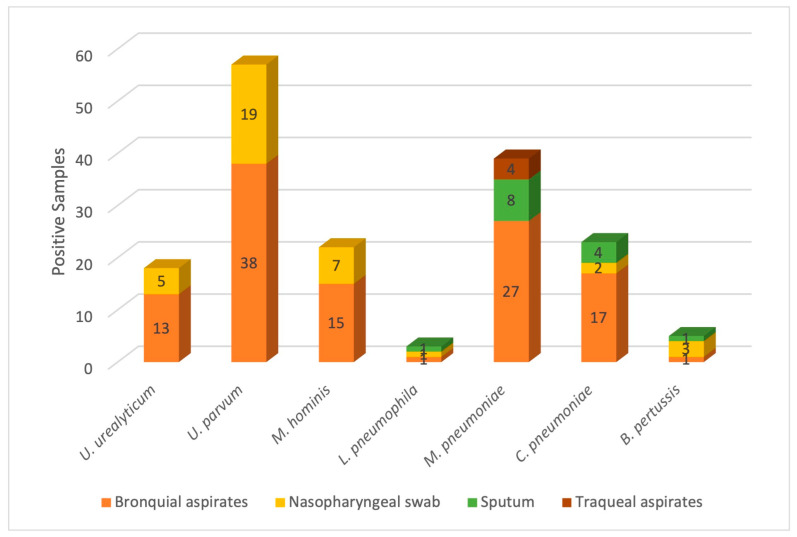
Absolute frequency of atypical pathogens. The figure shows the number of cases for which atypical bacteria were identified, as well as the type of sample in which identification was performed. As observed, bronchial aspirates were found to exhibit the highest positive sample rates.

**Figure 2 microorganisms-12-02328-f002:**
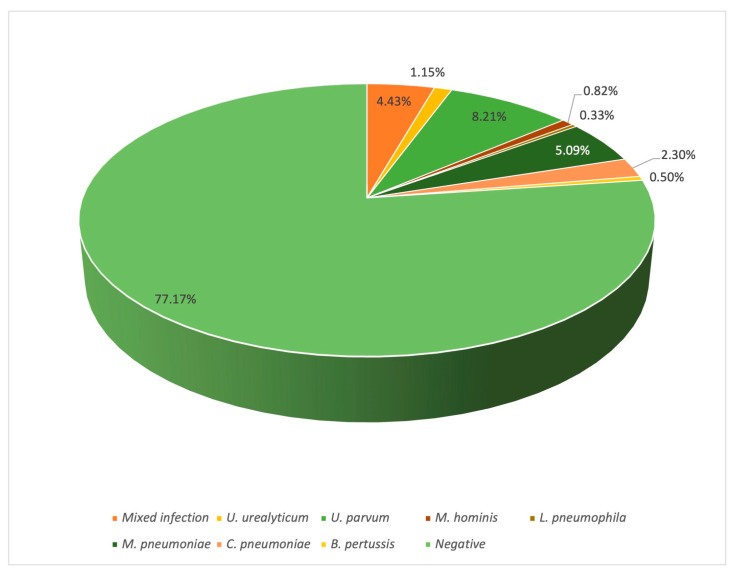
Relative frequency of pathogens causing AP. As observed in the plot, at least one atypical bacterium was identified in 22.83% of the sample. The percentages for each atypical bacterium considered in this study are represented by different colours.

**Figure 3 microorganisms-12-02328-f003:**
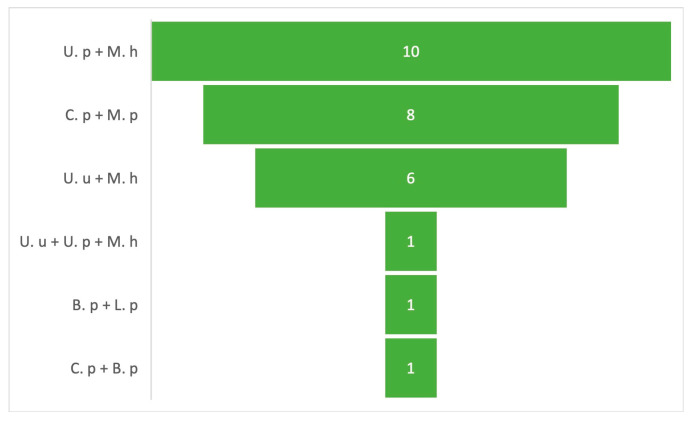
Clinical cases of mixed infections. The figure shows the frequency of mixed infections identified in the study. U. u: *U. urealyticum*, U. p: *U. parvum*, M. h: *M. hominis*, M. p: *M. pneumoniae*, C. p: *C. pneumoniae*, L. p: *L. pneumophila*, and B. p: *B. pertussis*.

**Figure 4 microorganisms-12-02328-f004:**
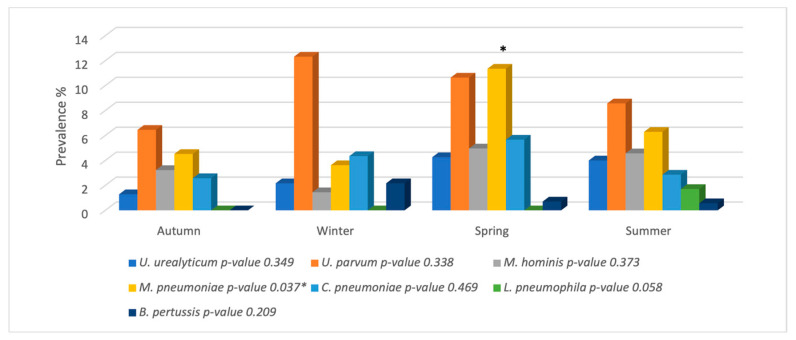
Prevalence of atypical pathogens according to the four seasons. *p*-value < 0.05: statistically significant difference between the seasons and the presence of each pathogen.

**Figure 5 microorganisms-12-02328-f005:**
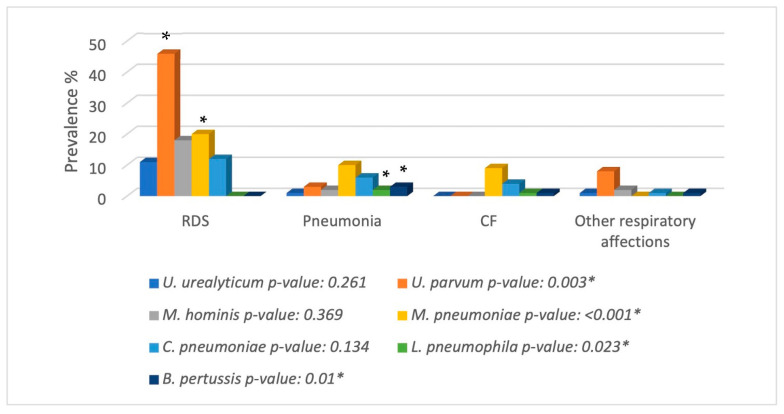
Prevalence of atypical pathogens according to the types of respiratory diseases. The figure shows that *U. parvum* and *M. pneumoniae* are more frequent in patients with RDS. All bacteria studied in this research were present in pneumonia, contrary to their presence in cystic fibrosis and other respiratory diseases. CF: cystic fibrosis.

**Figure 6 microorganisms-12-02328-f006:**
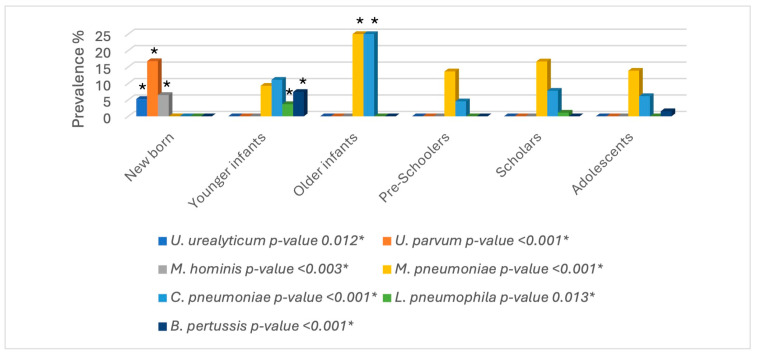
Prevalence of atypical pathogens according to paediatric group. *p*-value < 0.05: statistically significant difference between the paediatric groups and the presence of each pathogen.

**Table 1 microorganisms-12-02328-t001:** Sample classification.

Total Samples	Age Group *	Diagnosis/Number of Cases
Entire study n = 609	Newborns n = 340	RDS/n = 278
		Pneumonia, n = 24
		Other respiratory diseases, n = 38
	Younger infants n = 54	RDS/n = 25
		Pneumonia, n = 27
		Other respiratory diseases, n = 2
	Older infants n = 16	RDS/n = 3
		Pneumonia n = 8
		Other respiratory diseases, n = 5
	Preschoolers n = 44	RDS, n = 26
		Pneumonia, n = 12
		Cystic fibrosis, n = 4
		Other respiratory diseases, n = 2
	Scholars n = 90	RDS, n = 42
		Pneumonia, n = 23
		Cystic fibrosis, n = 25
	Adolescents n = 65	RDS, n = 42
		Pneumonia, n = 5
		Cystic fibrosis, n = 18

* The age group was identified according to NOM-008-SSA2-1993, Mexico. Newborns: <28 days old; younger infants: <1 year old; older infants: one year to one year and 11 months old; preschoolers: two years to four years old; scholars: five years to nine years old; adolescents: ten years to nineteen years old. RDS: respiratory distress syndrome. Other respiratory diseases: altered pulmonary transition, transitory tachypnoea, respiratory, insufficiency, respiratory alteration, asthma, bronchitis, and whooping cough syndrome.

**Table 2 microorganisms-12-02328-t002:** Characteristics of the primers.

Organism	Gen	Gen Product	Direction	Primer	Size (pb)
*L. pneumophila*	*Mip*	Macrophage infectivity enhancer	F	AGCATTGGTGCCGATTTGGGGA	101
			R	TGAGCGCCACTCATAGCGTCTT	
*Homo sapiens sapiens*	*HBB*	Human beta globin	F	ACCCTTAGGCTGCTGGTGGT	152
			R	AGGTGAGCCAGGCCATCACT	
*M. pneumoniae*	*P1*	Adhesine	F	TGGCTTGTGGGGCAGTTACCAA	275
			R	ACTGGGTGGGTAAACAAGCGGT	
*C. pneumoniae*	*ompA*	Outer membrane protein	F	ACTGGATCCGCTGCTGCAAACT	339
			R	ACCGCATTCCCATAAGGCTCCA	
*U. parvum*	*ure*	Urease	F	ACTGAGACACGGCCCATACT	244
			R	TAAATCCGGATAACGCTTGC	
*B. pertussis*	*BP3312*	Transposase	F	CTGCTTGACCGCAGTTCTC	518
			R	CGAAAGGTCCTGGAGACGTA	
*U. urealyticum*	*ure*	Urease	F	TGCTGCGCTAACGCAAAACTGT	679
			R	TCCCCTTGGGCAACGTCGAT	
*M. hominis*	*16S*	rRNA	F	AACCCCGGCTCGCTTTGGAT	753
			R	ACCCGAGAACGTATTCACCGCA	

**Table 3 microorganisms-12-02328-t003:** Clinical variables of the mother and the new-born patients, and its relationship with RDS.

Clinical Variables of the Mother		RDS		Total	*p*-Value
		No	Yes		
Type of birth	Caesarean section	63 (88.73%)	179 (89.05%)	242 (88.97%)	0.941
	Eutocic	8 (11.27%)	22 (10.95%)	30 (11.03%)	
Infection by mycoplasma	No	50 (70.42%)	160 (79.60%)	210 (77.21%)	0.113
	Yes	21 (29.58%)	41 (20.40%)	62 (22.79%)	
CCV	No	49 (69.01%)	165 (82.09%)	214 (78.68%)	0.021 *
	Yes	22 (30.99%)	36 (17.91%)	58 (21.32%)	
PROM	No	56 (78.87%)	150 (74.63%)	206 (75.74%)	0.473
	Yes	15 (21.13%)	51 (25.37%)	66 (24.26%)	
Age	Adolescent	4 (5.63%)	12 (5.97%)	16 (5.88%)	0.96
	Reproductive	60 (84.51%)	167 (83.08%)	227 (83.46%)	
	Risk	7 (9.86%)	22 (10.95%)	29 (10.66%)	
Clinical variables of the new-born					
Infection by mycoplasma	No	53 (74.65%)	152 (75.62%)	205 (75.37%)	0.87
	Yes	18 (25.35%)	49 (24.38%)	67 (24.63%)	
Sex	Female	28 (39.44%)	91 (45.27%)	119 (43.75%)	0.394
	Male	43 (60.56%)	110 (54.73%)	153 (56.25%)	
Weight	Low weight	51 (71.83%)	186 (92.54%)	237 (87.13%)	<0.001 *
	Normal weight	20 (28.17%)	15 (7.46%)	35 (12.87%)	
Respiratory support	Non-invasive	45 (63.38%)	109 (54.23%)	154 (56.62%)	<0.001 *
	IT/OT	16 (22.54%)	88 (43.78%)	104 (38.24%)	
	Not necessary	10 (14.08%)	4 (1.99%)	14 (5.15%)	
Prematurity	No	17 (23.94%)	14 (6.97%)	31 (11.4%)	<0.001 *
	Yes	54 (76.06%)	187 (93.03%)	241 (88.6%)	

RDS: Respiratory distress syndrome CCV: Cervicovaginitis; PROM: Premature rupture of membranes; IT/OT: orotracheal intubation; * *p*-value < 0.05.

**Table 4 microorganisms-12-02328-t004:** Adjusted prevalence between RDS and independent variables.

Variable		RDS RR	CI_95%_	*p*-Value
		SDR		
Sex	Male	-	-	-
	Female	1.087	0.953–1.241	0.214
Respiratory support	No	-	-	-
	Yes	2.411 *	1.092–5.320	0.0294 **
Intubation	No	-	-	-
	Yes	2.750 *	1.245–6.076	0.0124 **
Low weight	No	-	-	-
	Yes	1.689 *	1.153–2.474	0.00712 **
Infection by atypical bacteria	No	-	-	-
	Yes	0.845	0.703–1.017	0.0750
CVV	No	-	-	-
	Yes	0.823	0.672–1.007	0.0585
Prematurity	No	-	-	-
	Si	3.066 *	1.229–7.649	0.016 **

* = risk factors for development of RDS. CVV= cervicovaginitis. RR= relative risk. CI_95%_= confidence interval at 95%. ** *p*-value < 0.05.

**Table 5 microorganisms-12-02328-t005:** Relationship between the severity of RDS and newborn infection caused by atypical bacteria.

		RDS Severity			Total
Infection		1	2	3	
Negative	n	119	19	14	152
	%	78.29	67.86	66.67	75.62
Positive	n	33	9	7	49
	%	21.71	32.14	33.33	24.38
Total	n	152	28	21	201
	%	100	100	100	100
					*p*-value: 0.299

*p*-value < 0.05. 1: Mild. 2: Moderate. 3: Severe.

## Data Availability

The original contributions presented in the study are included in the article, further inquiries can be directed to the corresponding author.

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
