# Peer review of "Prevalence of Atypical Bacteria in Patients from Different Paediatric Age Groups Diagnosed with a Respiratory Disease"

_microorganisms, 2024, doi:10.3390/microorganisms12112328_

Round 1
Reviewer 1 Report
Comments and Suggestions for Authors
This is an study on the prevalence of atypical bacteria in patients with respiratory problems. The first thing for the authors is define atypical bacteria in that they are included in the study other bacteria such as Legionella or Bordetella that are not generally consider atypical bacteria. Another important point is that they use PCR to analyze samples and would be an interesting point but there are not comments on the results obtained with this method in relation to other identification systems and most important to the clinical data of the patients with positive PCR. The Discussion is very confusing, includes data from other sources with also too much details that are not important for the study; it has to be completely re-writted.
Some concerns:
lines 114-115: already said in line105
line 152: writte more clear
line 177: a total of 139.....samples?
lines 181-183: please explain
line 208: looking at the figure the statistically bsinificant is winter, not spring
Figure 5: the asterisk for U. parvum is missed
Figure 5; I cannot understand the stat. signf for pneumonia
line 223: more frequent in RDS is U. parvum but also M. pneumoniae
lines 245-256: info not needed, just highlight the factors favouring RDS
There is a problem with the tables: I cannot find Table 3
Table 5 (should be 4?) how the RDS severendess has been measured?
line 294: which several reasons?
lines 295-298: writte more clear
lines 306-311: it is not needed, the study is dealing only with atypical bacteria
line 315: please explain 6-40%
lines 317-322: delete
Comments on the Quality of English LanguageEnglish of medium quality, please revise
Author Response
In attention of your comments, I would like to respectfully respond.
Comments 1: This is an study on the prevalence of atypical bacteria in patients with respiratory problems. The first thing for the authors is define atypical bacteria in that they are included in the study other bacteria such as Legionella or Bordetella that are not generally consider atypical bacteria. Another important point is that they use PCR to analyze samples and would be an interesting point but there are not comments on the results obtained with this method in relation to other identification systems and most important to the clinical data of the patients with positive PCR. The Discussion is very confusing, includes data from other sources with also too much details that are not important for the study; it has to be completely re-writted.
Response 1: Indeed, as you say, Legionella and Bordetella are not atypical, however, they are relevant due to their relationship with respiratory tract infection and given their low frequency, the few cases that are presented are sometimes omitted. In this study, we consider them atypical due to their low frequency. In lines 47-49, we add information to the reader to clarify this point.
Regarding the clinical data, of the 140 patients with positive PCR, we had access to information from 97 patients. In the text we mention that we only had access to 272 newborn records and the data are those presented in table 3, which are discussed. Detection was not performed by other methodologies, so we cannot follow your recommendation.
Regarding the discussion, we restructured the ideas, so that there is a better understanding, ideas were added to enrich the discussion and some others that were out of context or that were repeated were deleted, hoping that with all these modifications, an enriching discussion has been achieved.
Some concerns:
lines 114-115: already said in line105:
Response 2: Thanks for the comment, we have removed the duplicate information. Only line 119 remains.
line 152: writte more clear.
Response 3: The explanation was detailed. Lines 168-172
line 177: a total of 139.....samples?
Response 4: The number of positive samples is not equal to the number of pathogens detected, since more than one was identified in some samples. Line 196-201. We rewrote this part to make it more understandable, graphs 1 and 2 were also modified.
lines 181-183: please explain
Response 5: The explanation was detailed in lines 202 to 205
line 208: Looking at the figure the statistically significant is winter, not spring.
Response 6: We appreciate the comment. The graph has already been corrected, it is on line 232, what was wrong was the graph, not the text.
Figure 5: the asterisk for U. parvum is missed.
Response 7: he asterisk was placed, thank you. Between Line 232
Figure 5; I cannot understand the stat. sign for pneumonia.
Response 8: Since there are few positive B. pertussis and L. pneumophila events and they are only cases of pneumonia, there is statistical significance when compared to other pathologies. Line 243
line 223: more frequent in RDS is U. parvum but also M. pneumoniae.
Response 9: Thanks for the comment, it has already been mentioned. Line 244
lines 245-256: info not needed, just highlight the factors favoring RDS.
Response 10: The information suggested was removed, since the tables contain the relevant statistical results, to see if there are or not, and to be less repetitive. We only left the classification of weight and prematurity, information that we consider important so that the reader knows why we consider the newborn to be premature and underweight. Lines 267-272
There is a problem with the tables: I cannot find Table 3.
Response 11: Sorry, the tables were numbered incorrectly, the correction was made. There are 4 tables in total. Line 287
Table 5 (should be 4?) how the RDS severeness has been measured?
Response 12: Table 5 is correct, it does not exist, it should be 4
Response 13: The correction was made. The severity is measured by the amount of oxygen in the blood and corresponds to; 1 mild RDS; 2 moderate RDS; 3 severe RDS, line 265-266. We added the meaning of RDS in the table: I, II and III. Between lines 297 and 298
line 294: which several reasons?
Response 14: By restructuring the discussion, this ambiguous part was eliminated.
lines 295-298: writte more clear.
Response 15: Same case, this idea was deleted since it was not an idea in line with the results.
lines 306-311: it is not needed; the study is dealing only with atypical bacteria.
Response 16: The idea was reduced, removing the examples of viruses so as not to stray from the main theme. Line 308-309
line 315: please explain 6-40%.
Response 17: We made a mistake in the wording. What we mean is that worldwide the prevalence of atypical bacteria ranges from 6 to 40%, depending on the country. In our study, the prevalence is 22.83%, a figure that is within the range reported worldwide. Lines 303-304
lines 317-322: delete.
We deleted the line as suggested because the data we were comparing with were from adult patients, different from our study group.
Reviewer 2 Report
Comments and Suggestions for Authors
The number of children who were introduced in the study was significant.
1. Even if in the bibliography and in the text there are references related to the age groups of the patients. We consider necessary the age to be specified between brackets for the readers that are less experienced with that. For example microbiologists and infectious diseases specialists.
2. For the clinicians who are reading this article, it would be interesting to know the proper antibiotic therapy, respectively first line antibiotics in those cases.
3.The experience of the authors of the article can not be generalized because as it was presented in the article, both Garcia and those from Iran did not have data similar to those found by the authors. Therefore, references should be made to the area to which the conclusion of this study may be applicable, so they are relevant to the connection between the possible epidemic of infections in women, at least for the newborn age group.
Author Response
In attention of your comments, I would like to respectfully respond.
The number of children who were introduced in the study was significant.
Comment 1: Even if in the bibliography and in the text, there are references related to the age groups of the patients. We consider necessary the age to be specified between brackets for the readers that are less experienced with that. For example microbiologists and infectiousdiseases specialists.
Response 1: Thank you for this observation, we placed the age range for each age group in table 1, lines 132-134
Comment 2: For the clinicians who are reading this article, it would be interesting to know the proper antibiotic therapy, respectively first line antibiotics in those cases.
Response 2: We include in the introduction some lines (87-93) referring to antibiotic treatment against these atypical agents.
Comment 3: The experience of the authors of the article cannot be generalized because as it was presented in the article, both Garcia and those from Iran did not have data like those found by the authors. Therefore, references should be made to the area to which the conclusion of this study may be applicable, so they are relevant to the connection between the possible epidemic of infections in women, at least for the newborn age group.
Response 3: We consider contrasting Garcia's results against ours because they also use PCR, and it is a similar study, we leave this written in the discussion line 368, considering his observation, which we appreciate, in the conclusions we add an important idea about the prevalence in different geographic areas given the heterogeneity of the percentages in different articles, lines 443-444
Round 2
Reviewer 1 Report
Comments and Suggestions for Authors
Authors responded adequatelly to the reviewer comments and the required modifications/corrections have been done